# The Examination of Restrained Joints Created in the Process of Multi-Material FFF Additive Manufacturing Technology

**DOI:** 10.3390/ma13040903

**Published:** 2020-02-18

**Authors:** Janusz Kluczyński, Lucjan Śnieżek, Alexander Kravcov, Krzysztof Grzelak, Pavel Svoboda, Ireneusz Szachogłuchowicz, Ondřej Franek, Nikolaj Morozov, Janusz Torzewski, Petr Kubeček

**Affiliations:** 1Institute of Robots and Machines Design, Faculty of Mechanical Engineering, Military University of Technology, 2 Kaliskiego St., 00-908 Warsaw, Poland; lucjan.sniezek@wat.edu.pl (L.Ś.); krzysztof.grzelak@wat.edu.pl (K.G.); ireneusz.szachogluchowicz@wat.edu.pl (I.S.); janusz.torzewski@wat.edu.pl (J.T.); 2Faculty of Civil Engineering, Czech Technical University in Prague, Thákurova 7/2077, 166 29 Prague 6-Dejvice, Czech Republic; alexandr.kravcov@fsv.cvut.cz (A.K.); pavel.svoboda@fsv.cvut.cz (P.S.); ondrej.franek@fsv.cvut.cz (O.F.); morozni1@fsv.cvut.cz (N.M.); petr.kubecek@fsv.cvut.cz (P.K.)

**Keywords:** Additive manufacturing, FFF technology, laser amplified ultrasonography, tensile testing

## Abstract

The paper is focused on the examination of the internal quality of joints created in a multi-material additive manufacturing process. The main part of the work focuses on experimental production and non-destructive testing of restrained joints of modified PLA (polylactic acid) and ABS (Acrylonitrile butadiene styrene) three-dimensional (3D)-printed on RepRap 3D device that works on the “open source” principle. The article presents the outcomes of a non-destructive materials test in the form of the data from the Laser Amplified Ultrasonography, microscopic observations of the joints area and tensile tests of the specially designed samples. The samples with designed joints were additively manufactured of two materials: Specially blended PLA (Market name—PLA Tough) and conventionally made ABS. The tests are mainly focused on the determination of the quality of material connection in the joints area. Based on the results obtained, the samples made of two materials were compared in the end to establish which produced material joint is stronger and have a lower amount of defects.

## 1. Introduction

Due to the fact that polylactic acid (PLA) is environmentally friendly in accordance with its potential biological diversity, taking into account its natural origin, the material has become one of the most promising biopolymers. At the same time, PLA has high strength, stiffness, resistance to fats and oils [1,2]. However, PLA has significant drawbacks that limit its use in various applications: Low impact resistance, short elongation, and low-temperature resistance. The stress at break is usually less than 10%, and the impact strength is below 2.5 kJ/m^2^ [3,4,5]. In order to compensate for the shortcomings of PLA, acrylonitrile butadiene styrene (ABS) stands out among the hardening polymers due to its high elongation during tensile tests, good durability and ease of extrusion and molding. An additional advantage of additively manufactured elements using styrene-based materials is the possibility of numerical analysis conduction. Kucewicz et al. used that kind of material (ABSPlus) to produce three-dimensional (3D) lattice-structured elements and prepare their numerical models that were able to be validated during the experimental phase [6,7].

Both PLA and ABS do not mix, their mixture leads to a hybrid material, which leads to weak mechanical properties, as well as unstable morphology [8,9,10,11]. The above-mentioned materials have their limitations. PLA is characterized by a very low plastification temperature and a lack of UV resistance. ABS has low moisture resistance, and in addition to 3D printing solutions, it has a significant shrinkage which makes it difficult to use—on manufactured parts could appear cracks and delaminations. Regarding the above-mentioned restrictions, it is necessary to provide the best possible additive manufacturing technology and select proper materials for their usage conditions.

Nowadays there are a many publications connected to the additive manufacturing of different material types [12]. Also, in this type of manufacturing technology, there is a great need for composite materials production. In the available research results, there is a low amount of publication where it was taken into account multi-material additive manufacturing processes. Shin et al. [13] pointed out the most significant group of multi-material usage are applications connected with PolyJet additive manufacturing technology which is based on production using resins. Basing on resin usage, Hardin et al. [14] used multi-material inks to reach special properties of the produced elements for medical applications. A combination of the liquidized materials, like resins, seems to be easy-to-obtain in comparison to extruded, plasticized thermoplastic materials. Obviously, there are available technologies using plastics extrusion with the possibility of different material applications [15,16]. There are also applications using some fiber-fitting methods. Goh et al. [17] tested fiber-reinforced, 3D printed materials to reach very good mechanical properties of the manufactured materials. There are also available some conventional solution for obtaining composite materials which using plastic-welding or gluing. There are also some results connected to more advanced conventional methods of materials joining. Singh et al. [18] used friction welding to combine polymers with the additional metal powder reinforcement. The last group of some advanced, conventional methods is the gluing method using different types of adhesives or joints geometries. Smutek et al. [19] used especially prepared epoxy glue for joining the 3D printed polymers to reach better mechanical properties of designed joints.

The other way is to prepare the material joint using multi-extrusion 3D print in the FDM (Fused Deposition Modelling) or FFF (Fused Filament Fabrication) technology. Lopes et al. [20] used a modified 3D printing device to obtain elements composed of two types of thermoplastic materials that were manufactured directly in a single process, without any post-processing. The main issue of the prepared joints was the low mechanical properties of produced joints.

Basing on the last five years of research, the most significant article groups, connected with plastics joints, are focused on composites with the addition of polymers [21,22,23,24]. A significant part is taking into account conventional material joining type. More advanced technologies were shown in Eslami et al. research [24], where it had been used friction stir welding technology of polymers. A small amount of that kind of research and lack of new solutions for plastics was the main background for investigating a patented technology (Patent application number: P432635), and the tests are explained in this article.

In this work, the effectiveness and quality of the joints, using two materials, were taken into account. To reach very good joint mechanical properties it had been designed a proper additive manufacturing FFF process and special joint geometry. The prepared joints were tested using a non-destructive laser amplified ultrasonic method, supplemented by the microscopic observations of the joints before and after tensile tests. The research allowed the mixed material properties to be obtained which could be used in many different industrial applications. Additionally, polymer-joining technology is the alternative for conventional joining techniques (glue connections, screw connections, etc.). The research shows that developed connection geometry and manufacturing technology could be helpful for reaching polymer joints characterized by better mechanical properties.

## 2. Experimental

Fused Filament Fabrication (FFF) fabrication (Figure 1) is one of the most popular additive manufacturing technology, to ensure a low price for the devices, as well as the materials for the manufacturing process.

The FFF 3D print technology allows for obtaining elements with very complex geometry. The geometry of the manufactured elements are based on the data forms of the 3D CAD data. It is not necessary to use the 2D drawings for the process preparation of this type of data. During this process, the plastic wire is plastified in the heated nozzle. After the extrusion from the nozzle, plastified plastic goes to the building platform where there is the final shape construction. After the final creation of each layer of material, the building platform is lowered to let another layer creation. The process is repeated to reach the final shape of the element. Printing paths were designed with raster angles of +45°/−45° for each deposited layer, as depicted in Figure 2.

For the test, it was used the PLA Tough™ and ABS Smart™ material provided by the Spectrum Filaments Ltd. Company (Pęcice, Poland). The PLA Tough™ is specially blended PLA (PolyLactic Acid) which makes the material more thermal-resistant and improved from a mechanical properties point of view. The chemical composition provided by the material manufacturer of used materials was shown in Table 1.

To reach desirable elements properties, the following process parameters were designed:Hotend temperature (for both materials): 255 °C,Heatbed temperature: 75 °C,Layer thickness: 0.2 mm,Infill: 100%,Part cooling intensity: 40%,Printing speed: 50 mm/s,Nozzle diameter: 0.4 mm,Distance between extruded paths: 0.36 mm.

To evaluate mechanical properties, dog-bone shaped parts were fabricated by the FFF additive technology printer according to ASTM D638: “*Standard Test Method for Tensile Properties of Plastics*”. For each test, five same samples were prepared to make sure the obtained values of designed joints mechanical properties reliable. The geometry of designed samples with two different joints with the reference monolithic sample was shown in Figure 3.

To obtain the proper restraint joints it was used a smooth change of the material during the process. For each layer in samples with joints, it was put both materials basing the procedure shown in Figure 4.

To obtain a permanent joint of two materials, a four-way connector device (Prusa Research a.s., Prague, Czech Republic) was used in the Prusa Original MK2 with a Multi-Material Unit 1.0, which allowed for extruding two different materials, which were smoothly changed during the process. Using only a single printing nozzle required the same hot-end temperature application. Increased temperature resistance of PLA Tough™ prevented material degradation during extrusion in the temperature, which was higher above the processing temperature.

For the analysis, it was prepared two types of restraint joint geometry—the overlay connection (2 in Figure 4), and the pleated connection (3 in Figure 4), which were also shown and described in Figure 5.

In connection with the described experiment, the research was run based on the program shown in Figure 6.

## 3. Optoacoustic Equipment and Measuring Technique Results and Discussion

Laser ultrasonic structuroscopy (an original measurement tool built in Czech Technical University in Prague, Prague, Czech Republic), similar to traditional ultrasonic systems operating in echo-pulse mode, was used in this study [26,27]. The generation of short ultrasonic pulses of strictly controlled form occurs in an optoacoustic cell [28]. Figure 7 shows a schematic diagram of this cell for the diagnosis of materials with one-way access. A pulse generated by an Nd:YAG laser is transmitted to the front side of a special optical-acoustic generator (OAG) through a fiber optic cable, an optical beamforming system, and a transparent prism. OAG is a plane-parallel plate made of a special plastic that absorbs light [29]. The transparent prism is in acoustic contact with the OAS, at the same time as the sound-conducting channel of a broadband piezoelectric transducer. Access, on the one hand, and acoustic contact are provided by pressing the OAG plane to the front of the object with a thin layer of contact liquid [30].

The pulse energy enters the sample and is scattered by its heterogeneities and reflected from the backside of the sample. All scattered and reflected signals are recorded by the piezoelectric transducer and processed by a processing system [31].

Laser-ultrasonic structuroscopy was performed on samples AP–P1 with “smooth” joint and AP–Z1 with an “overlap” joint. The thickness of the products was measured before laser ultrasonic testing. The velocity of the longitudinal wave, along the direction perpendicular to the layers of the composite, was calculated by the thickness and difference of the arrival time of the signals reflected from the rear side of the object and the generator-object interface. The obtained elastic wave propagation velocities for the samples were: ABS sound velocity 2165 m/s, PLA sound velocity 2305 m/s. More than 10 first polymer layers are clearly distinguishable in the samples, which makes it possible to control the manufacturing technology of ABS and PLA products. The quality of the polymer layers was evaluated by images of the internal structure of the samples.

Figure 8 shows one of the obtained B-scans of the internal structure of the samples. The red region is the interference of the technological signal caused by the structure of the sensor itself: The difference in acoustic impedance of the optical-acoustic generator and the waveguide prism.

In Figure 8a,b, the yellow region indicates its rear side. The unevenness of the displayed backside, which is actually a flat surface, indicates that the binder is heterogeneous and has patches of less dense material. A change in the speed of elastic waves is directly related to a change in elastic properties. The difference from the average value was up to 100 m/s, which is 3%; with an accuracy of measuring the longitudinal wave velocity of 0.5%. Unevenness can also be due to defects in the internal structure or porosity.

During the study, defects were found in the form of delamination, pores, as well as areas with a violation of the geometry of the material laying. For a more visual representation, the most significant defects are displayed by red areas on 3D models, presented in Figure 9a,b.

## 4. Microstructural Analysis—Results and Discussion

Due to the exposed imperfections of the joints in the non-destructive laser amplified ultrasonography, the samples were prepared to microscopic observation by cutting the part of the sample in the joints area, grinding with abrasive papers (gradations: 80, 120, 320, 500, 800, 1000, 1500, 2000) and polished using 1 μm diamond paste. The microscopic observations were done on the Olympus LEXT 4100 confocal microscope (Olympus Corporation, Tokyo, Japan). The areas with imperfections (Figure 9) were also taken into account in the microscopic analysis and was shown in Figure 10.

The exposed imperfections during the non-destructive laser amplified ultrasonography was connected with the porosity and delaminations on the side part of both joints. The number of imperfections in the whole observed area was similar to the whole volume of the joint (shown in Figure 8). The microstructure of the joints in bigger magnifications obtained using Olympus LEXT 4100 confocal microscope for each sample sides is shown in Table 2.

As shown in Table 2, the borderline of each joint has not a significant imperfection in microscale. The exception involved some bigger imperfections (shown in Figure 9), which were present only in a small part of the whole joint area. Only one front view image of the sample for both connection types is related to the same used wavy shape and the same types of the image. As can be seen in the side view of the overlap connection the PLA material (dark color) came into the frayed structure in the ABS material. This phenomenon had a significant influence on the joint material properties which was a part of the material in the next chapter of this article. On the side view of the pleated connection, it is visible an area of both material penetration, where one material came into another material surface irregularities. In the front view of both designed connections, there is visible both materials overlapping, which was connected with lowering the distance between the extrusion path.

## 5. Tensile Test—Results and Discussion

The tensile tests were ran basing on the ASTM D638 standard. All tensile tests (results shown in Figure 11) were performed under room temperature conditions using a universal testing machine (Instron, Norwood, CO, USA) Instron 8802 with a quasi-static loading condition (5 mm/min). Additional microscopic observations (shown in Figure 12, Figure 13 and Figure 14) were made using an Olympus LEXT 4100 confocal microscope. The obtained material properties included tensile strength and elongation at break of the samples are presented in Table 3.

The monolithic PLA Tough™ had 18% higher tensile strength than ABS and 21% lower elongation at break, as demonstrated in the available literature. The first characteristic issue connected with the used joints is that it is possible to improve the mechanical properties of the element. It could be seen that the overlap connection between both parts, made of PLA, increased the specimen’s tensile strength by 2.3% with the increase of the strain at the break by 21%. Using a pleated joint in connection with both parts made of PLA (PTZ) increased the tensile strength by 14% with increasing the elongation of the material by 22%. The different phenomenon had been spotted in the joints of the elements made by connecting two parts made of ABS. Their elements with joints had tensile strength increased (5% for overlap connection—ABSN, and 2% for pleated connection—ABSZ) in both connection types of the elements made of ABS there is visible an elongation at break decreasing (4% for overlap connection—ABSN, and 11% for pleated connection—ABSZ). The results for the elements, made of two different materials using designed joints, were most important. The tensile test results for the samples with used joints of two different materials were shown on the chart in Figure 11 using dash-dotted curves.

Both elements with used joints had better tensile strength than monolithic elements made of ABS and higher strain at break than monolithic PLA Tough™ elements. Based on these results, there is a possibility of joining the polymers which cannot be blended and can maintain the advantages of both connected elements. Additionally, designed joints types allow for increasing the strength properties of the connected materials. This phenomenon is connected with the adhesive character of the designed joints which had a positive influence on the element’s strength properties (especially made of PLA Tought™).

The influence of additional adhesive character was able to notice the results from the digital image correlation (DIC) analysis of the elements deformations during tensile tests—Table 4. Local strain fields were acquired by Digital Image Correlation (DIC) using the Dantec Q-400 System. Before testing, the specimens were prepared by applying a random black speckle pattern, over the previously mat white painted surface, in order to enable strain data acquisition by DIC. From the DIC results, a series of strain field images were extracted to reveal the deformation evolution of the joint. Strain and displacement maps were created for each specimen using the Dantec Istra 4D software.

It could be seen that elements made of PLA have much bigger deformations distribution than elements made of ABS, where the material deforms only near the narrowness before the fracture. In the joint analysis, there could be noticed that in overlap joints there are much bigger deformations distributions in the measured area. It could be noticed that the joint is similarly deformed in both sides of the connection borders.

As could be seen on the chart (Figure 11) specimens with overlap joint had better strength properties than specimens with pleated joints (2% higher tensile strength and 8% higher elongation at break). This phenomenon is connected with the joint characteristic. In Figure 11 it was shown the fracture of the specimen with an overlap joint where it could be seen that specimen was damaged right behind the part made of PLA. Additionally, the elements manufactured with PLA-ABS overlap joint are characterized by positive deformations distribution during loading, where the joint “works” in its whole volume. In elements with that kind of joint, the material fracture was on the ABS-side where the fracture area has brittle characteristics with some areas of ductile cracking (Figure 12a). By comparing ABS with PLA, there was a different type of cracking mechanism than in the ABS elements. PLA fracture was characterized by typical brittle cracking (Figure 12b) with the presence of many longitudinal cracks (Figure 12c). The fracture of the overlap PLA-ABS joint is shown in Figure 12.

During the fracture analysis of the elements made of PLA and ABS connected using a pleated joint it had been noticed delamination of the joint on the PLA-ABS border and characteristic for the ABS brittle-ductile fracture in the rest of the connection, which was shown in Figure 13.

Based on the fracture analysis, two types of material delamination are seen. The first type was connected with the tensile strength of the ABS material, the second was connected with the adhesive connection between both materials.

The same—dual nature fracture had been spotted in overlap joints, where the joint was broken in the connection border. Better tensile properties of the overlap joint could be caused by the summation of both types of joint strength—the material strength and the adhesive connection strength. The fracture of the ABS material with the PLA particles was shown in Figure 14.

As shown in the research results, using these types of joints increases the element’s strength properties, which was able to observe during the fracture analysis. It can, therefore, be concluded that designed types of joints allow for improving the joining materials strength properties, where it is also possible to produce elements with different characteristics on each side of the joint in these elements.

## 6. Final Conclusions

Acrylonitrile butadiene styrene (ABS) and polylactic acid (PLA) specimens were created according to the ASTM standards D-638 to determine the tensile strength of two designed joints for those materials connection. Based on the research results, the following conclusions were found:The study revealed that the samples with an “overlap” joint have greater strength than specimens with a “smooth” joint. This is due to the larger contact area, plane-parallel connection geometry and double nature of the material strength at the joint borders.Contact laser ultrasound spectroscopy can be used to visualize the internal structure of composite materials. It is also possible to control the quality of special joints and detect internal defects to evaluate the number and thickness of layers of PLA and ABS plastics.The use of “shaped-adhesive” connections allows elements with different properties to be obtained, depending on the materials used.The developed method is a real alternative to glued joints with this type of material.The method is proper for the connection of all 3D-printable materials, including elastomers and materials with the addition of metal and ceramic powders.In relation to solutions in the literature, the investigated technology allows for better joint properties to be obtained than presented in the paper [3].To achieve a better performance of the obtained joints, heat treatment could be helpful and will be used in further research.The pleated connection areas are characterized by both materials (PLA and ABS) penetration, where one material came into another material surface irregularities. Investigated connection geometry and manufacturing techniques allow for materials overlapping which were connected with lowering the distance between the extrusion path.The overlap connection between both parts made of PLA increased the specimen’s tensile strength by 2.3% with the increase of the strain at the break by 21%.The pleated joint of both parts made of PLA (PTZ) allowed increasing the tensile strength by 14% with increasing the elongation of the material by 22%.In both connection types of ABS parts, its tensile strength increased (5% for overlap connection—ABSN, and 2% for pleated connection—ABSZ). At the same time, elongation at break decreased (4% for overlap connection—ABSN, and 11% for pleated connection—ABSZ).

## Figures and Tables

**Figure 1 materials-13-00903-f001:**
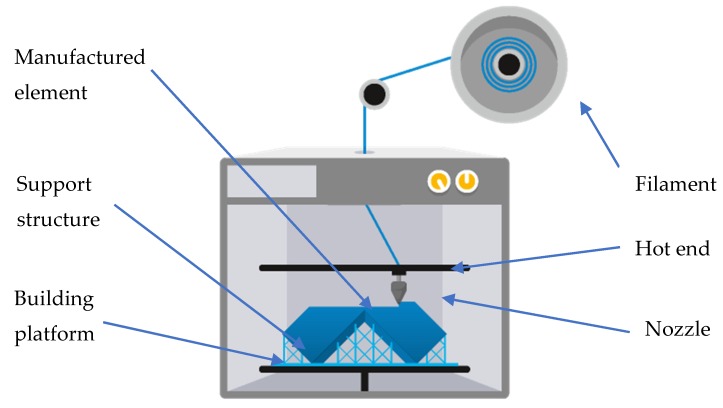
The FFF additive technology process scheme [25].

**Figure 2 materials-13-00903-f002:**
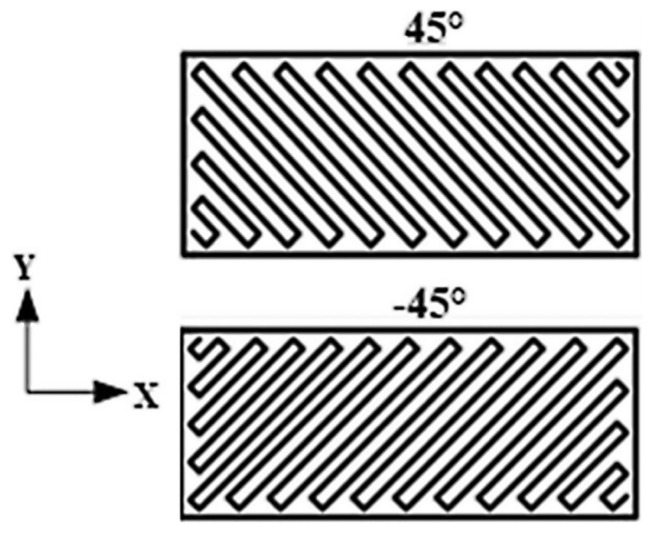
Layer deposition with different raster angles.

**Figure 3 materials-13-00903-f003:**
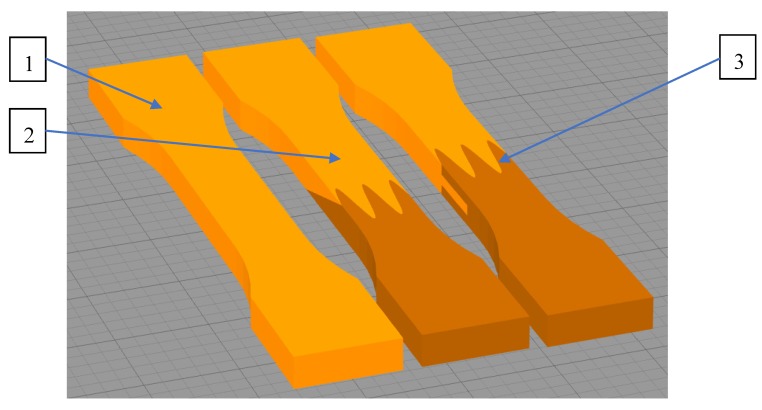
Designed samples: 1—the monolithic reference sample; 2—the overlap connection, 3—the pleated connection.

**Figure 4 materials-13-00903-f004:**
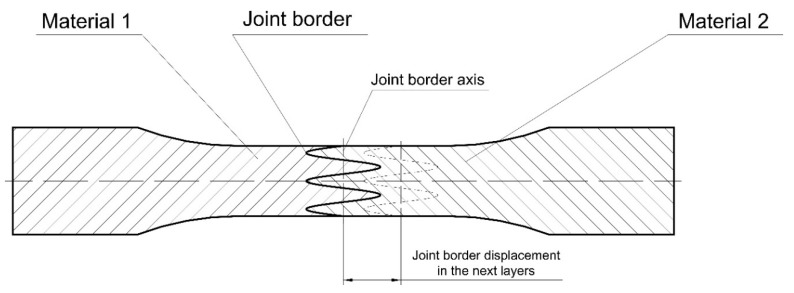
The joint manufacturing procedure for each layer.

**Figure 5 materials-13-00903-f005:**
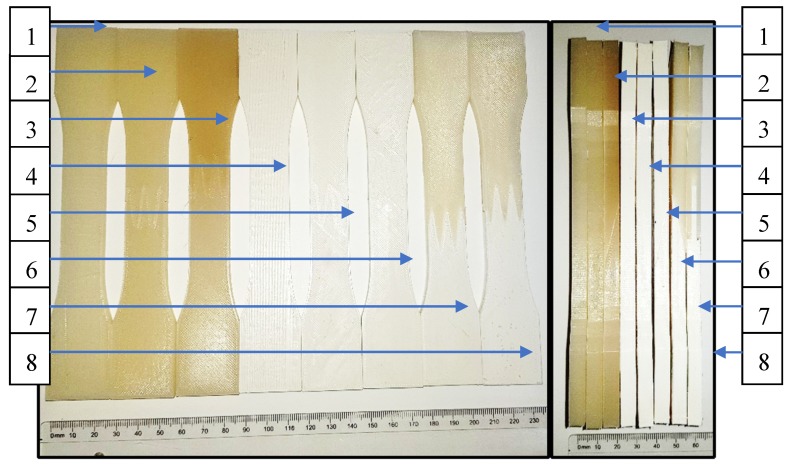
Additively manufactured samples: 1—Monolithic, PLA (PT); 2—Pleated connection, PLA (PTZ); 3—Overlap connection, PLA (PTN); 4—Monolithic, ABS (ABS); 5—Pleated connection, ABS (ABSZ); 6—Overlap connection, ABS (ABSN), 7—Overlap connection, PLA/ABS(PTABSZ); 8—Pleated connection, PLA/ABS (PTABSN).

**Figure 6 materials-13-00903-f006:**
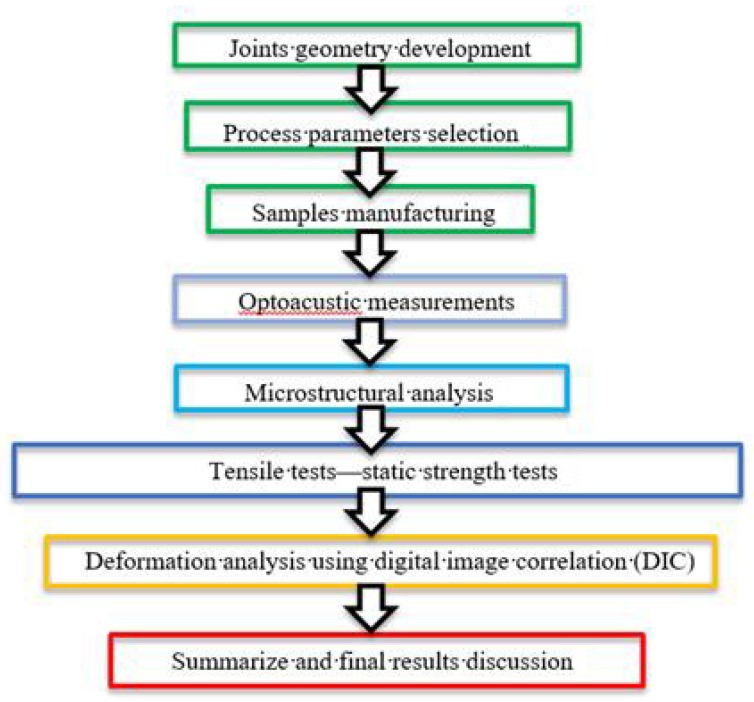
Illustration of the research methodology employed in this study.

**Figure 7 materials-13-00903-f007:**
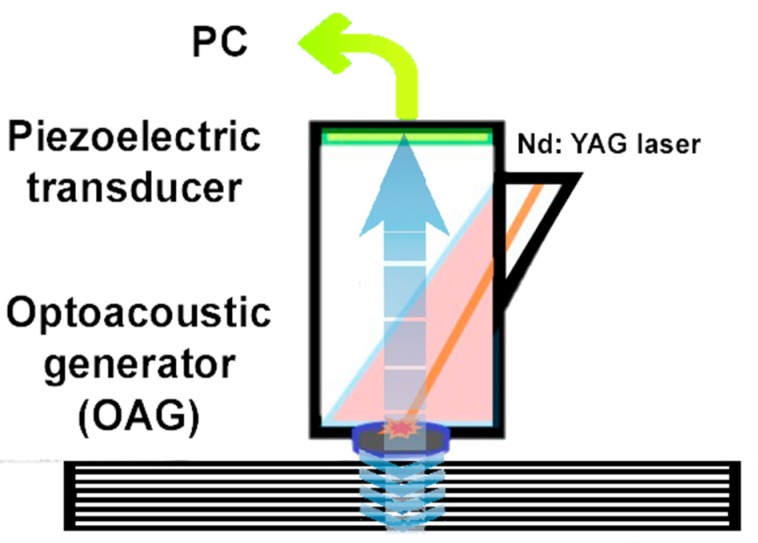
Schematic diagram of optoacoustic generator.

**Figure 8 materials-13-00903-f008:**
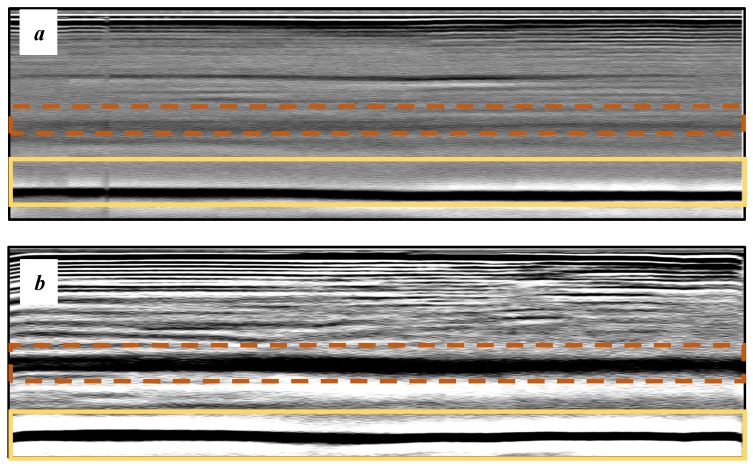
The internal structure of samples; (**a**) Pleated connection, PLA Though™, and ABS; (**b**) Overlap connection, PLA Though™, and ABS.

**Figure 9 materials-13-00903-f009:**
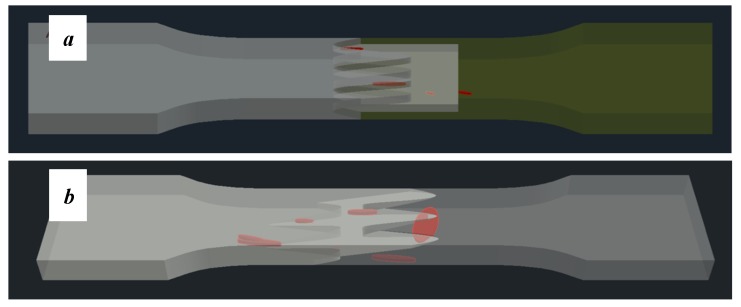
Three-dimensional (3D) model of samples with defective areas designation; (**a**) pleated connection, PLA Though™, and ABS; (**b**) overlap connection, PLA Though™, and ABS.

**Figure 10 materials-13-00903-f010:**
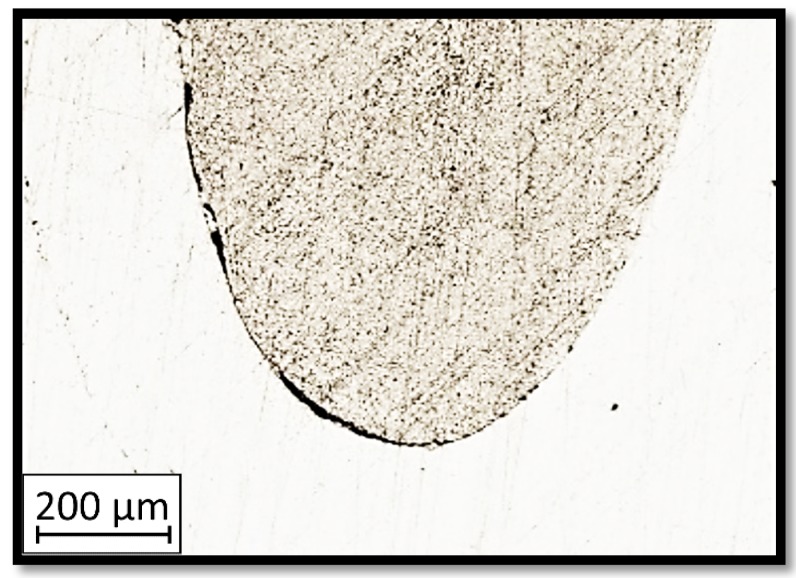
Imperfections (black areas) at the end of the wavy borderline of the joint.

**Figure 11 materials-13-00903-f011:**
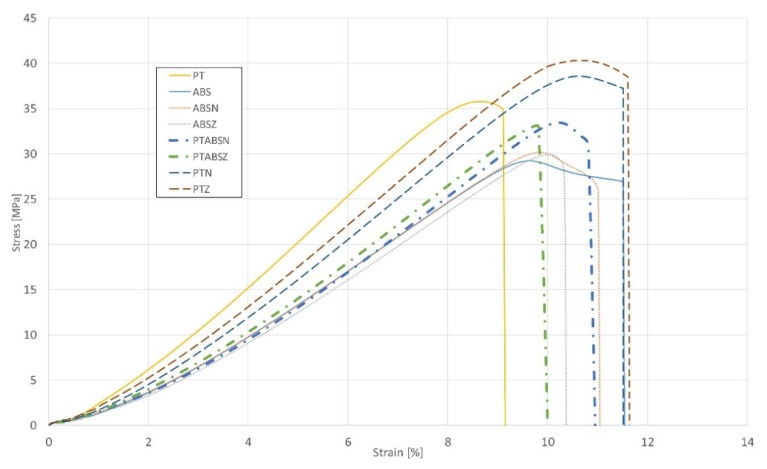
Comparison of stress-strain curves for all types of specimens tested.

**Figure 12 materials-13-00903-f012:**
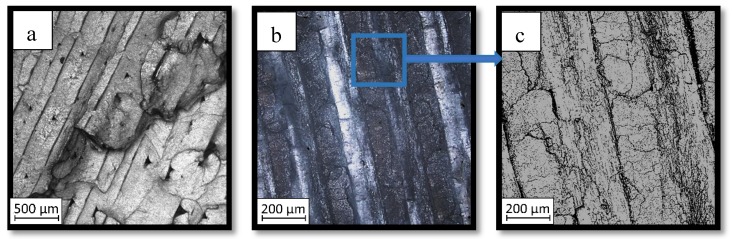
Fracture images for ABS (**a**) and PLA Tough™ (**b**,**c**).

**Figure 13 materials-13-00903-f013:**
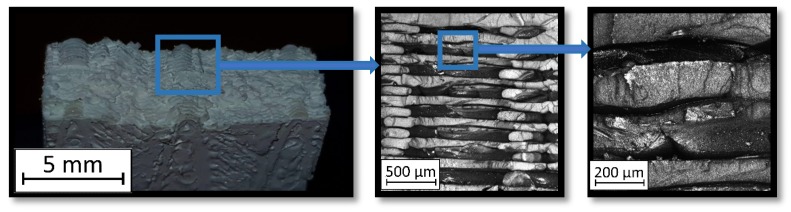
Fracture images of a pleated joint of PLA-ABS materials.

**Figure 14 materials-13-00903-f014:**
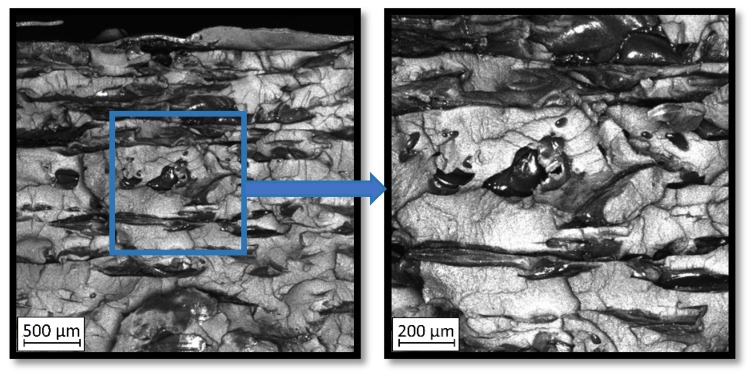
Fracture images of overlap joint of PLA-ABS materials.

**Table 1 materials-13-00903-t001:** Base chemical composition of used materials.

Material	PLA (wt.%)	Material	ABS (wt.%)
Polylactide resin	>75	Acrylonitrile	15–35
Magnesium Silicate	<25	Butadiene	5–30
Additional Blends	≈3	Styrene	40–60

**Table 2 materials-13-00903-t002:** The microstructure of the designed joints.

Overlap Connection—Side View	Pleated Connection—Side View	Pleated/Overlap Connection—Front View
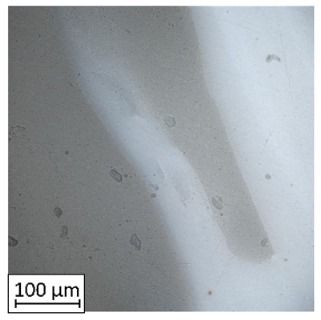		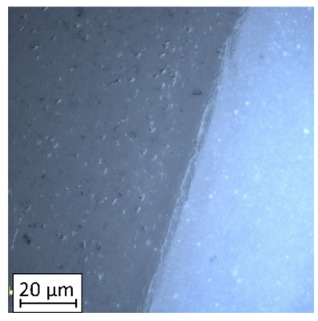

**Table 3 materials-13-00903-t003:** Mechanical properties of samples tested.

Tensile Tests Results	PT	ABS	PTN	PTZ	ABSN	ABSZ	PTABSZ	PTABSN
Ultimate stress (MPa)	35.8	29.4	38.6	41.3	30.8	29.9	32.7	33.4
Strain at break (%)	9.15	11.5	11.5	11.6	11.01	10.34	10.1	11.0

**Table 4 materials-13-00903-t004:** DIC strain distribution maps of the specimens during tensile tests.

Specimen Description	Strain at 0.2% of Elongation	Strain at R_m_	Strain at R_u_	Strain at Break	Scale
ABS	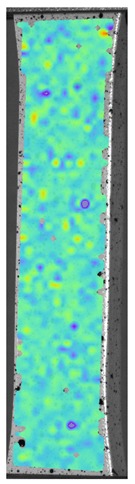	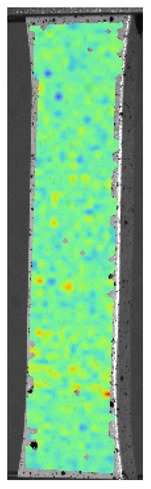	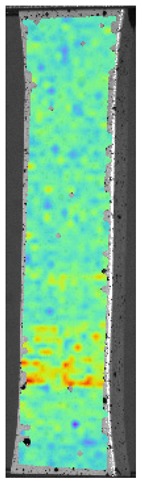	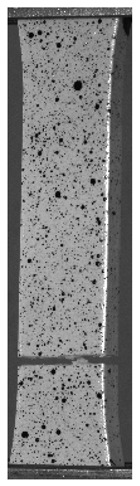	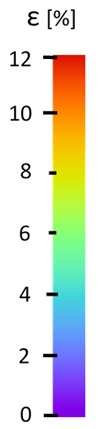
PT	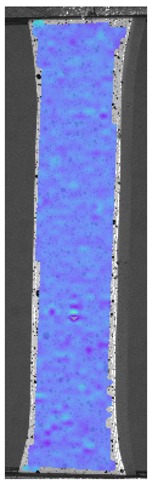	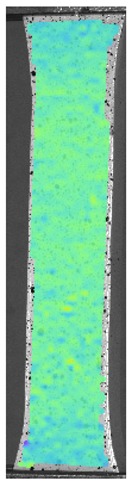	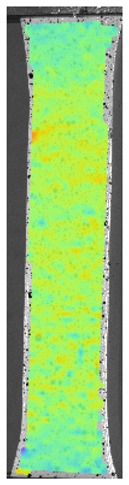	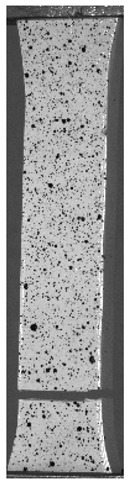	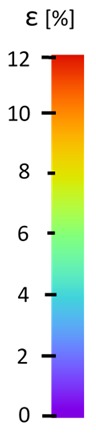
ASN	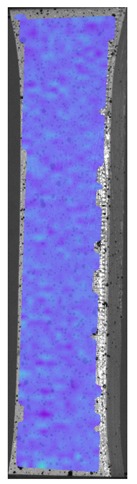	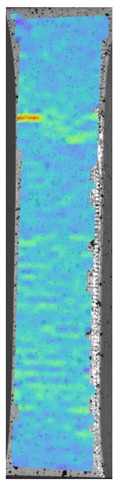	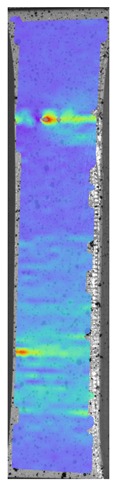	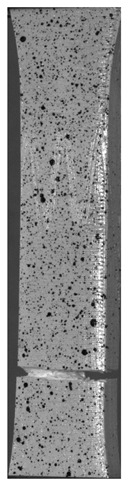	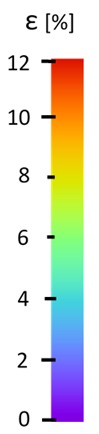
ASZ	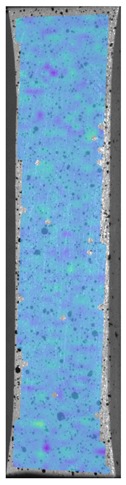	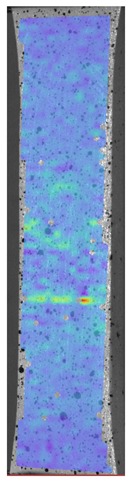	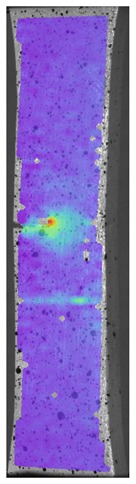	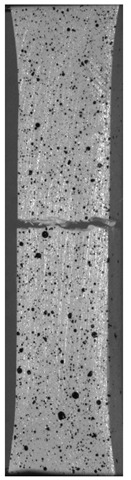	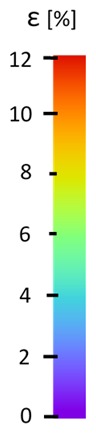
PTN	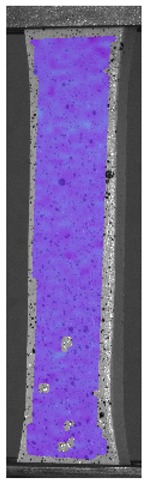	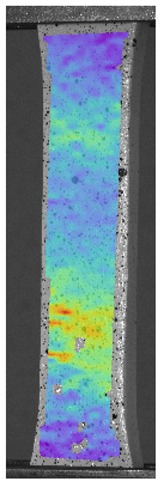	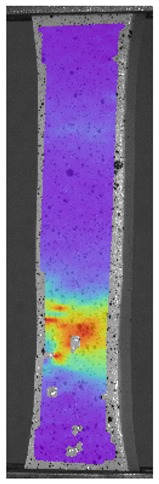	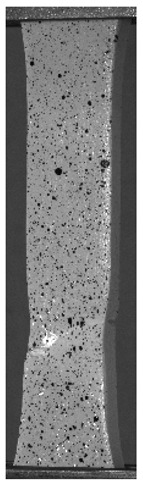	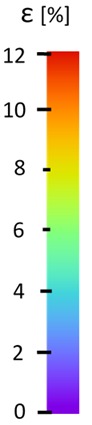
PTZ	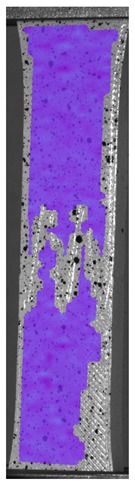	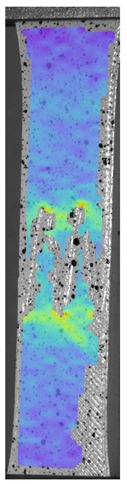	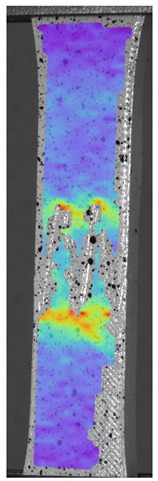	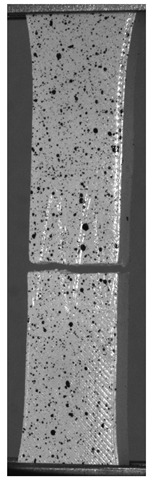	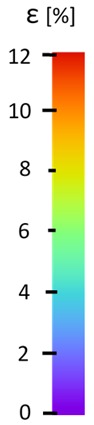
PTASN	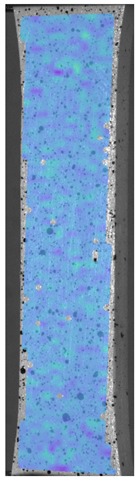	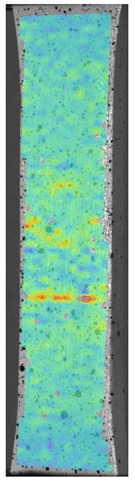	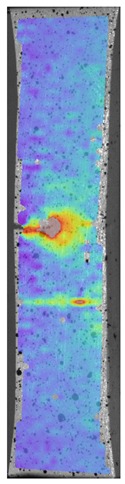	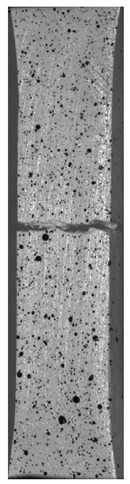	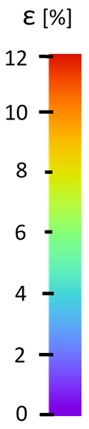
PTASZ	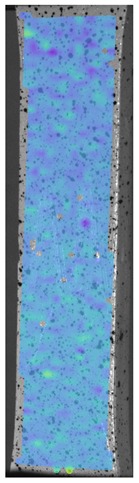	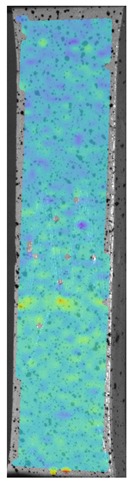	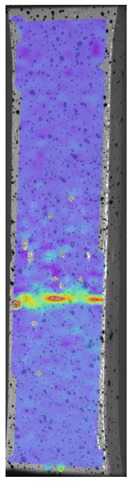	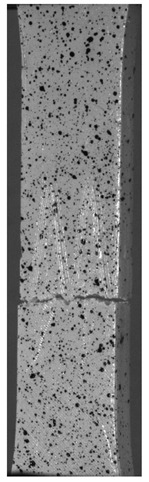	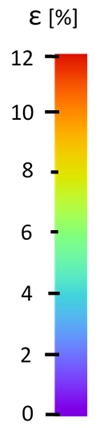

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
