# Peer review of "The Examination of Restrained Joints Created in the Process of Multi-Material FFF Additive Manufacturing Technology"

_materials, 2020, doi:10.3390/ma13040903_

Round 1
Reviewer 1 Report
This work is interesting, the manuscript is well represented and the results are comprehensive. In my opinion, it can be considered for publication pending the revision based on the following points:
What is the significance of the work is not clear? It should be added to the introduction part.
Add a schematic representation of the overall work.
Introduction needs revision with more up to date citations and details about the background of the present work.
Add all the details and specifications of chemicals used in the materials section.
What is the reason behind using PLA and ABS together? Explain.
What is the advantage of using contact Laser Amplified Ultrasonography? Explain.
Line 297-298: How the adhesive properties were measured?
The discussion could be more solid. So the results should be discussed with previously published literature.
Also, revise the manuscript carefully to reduce the typographical and linguistic errors in the manuscript.
Author Response
Dear Reviewer,
We would like to thank you for your revision and valuable comments.
In connection with your suggestion we have made the following corrections:
“What is the significance of the work is not clear? It should be added to the introduction part.”
COMMENTS:
We put the following text in the introduction and yellow-highlighted it:
Additionally, investigated polymers joining technology is a real alternative for conventional joining techniques (glue connections, screw connections, etc.). The research allows to state, that developed connection geometry and manufacturing technology could be helpful to reach polymer joints characterized by good mechanical properties.
“Add a schematic representation of the overall work.”
COMMENTS:
As an additional figure (fig 6) it was attached illustration of the research methodology employed in the study
“Introduction needs revision with more up to date citations and details about the background of the present work”
COMMENTS:
We have made the additional revision of background technologies connected with plastics joints which were included in the introduction and yellow-highlighted:
“Basing on the last five-years research work, the most significant articles groups connected with plastics joints are focused on composites with the addition of polymers [21-25]. A significant part is taking into account conventional material joining type. More advanced technologies were shown in Eslami et al. research [25], where it had been used friction stir welding technology of polymers. A small amount of that kind of research and lack of new solutions for plastics was the main background to investigate a patented technology (Patent application number: P432635) and made tests which were shown in this article.”
“Add all the details and specifications of chemicals used in the materials section.”
COMMENTS:
It was not used any chemicals during research. The only materials used were polymers which chemical composition we have added in an additional table (table 1).
“What is the reason behind using PLA and ABS together? Explain”
COMMENTS:
The main reason for using the mentioned materials was their popularity in the 3D printing market and the ability of manufacturing on generally available devices to make it possible to use even in basic solutions. Investigated joining technology could be used for other, less common materials (PET-G, TPU, TPE, PE, etc.).
”What is the advantage of using contact Laser Amplified Ultrasonography? Explain.”
COMMENTS:
It needs to identify the internal structures that are widely used by Nondestructive assessment (NDE). It is suitable for a wide variety of materials. There is a lot of non-destructive methods, such as THz spectroscopy, diagnostics by nonlinear optics, X-ray tomograph, and acoustics. Ultrasonic testing methods stand out from other methods due to simple and safe execution procedures and nevertheless a high rate of precision. However, there are some disadvantages. The main disadvantage deals with the usage of common piezoelectric transducers. These points means disability measurement of actual size and origin of the defects, complexity for investigating heterogeneous materials, for instance, coarse-grained metals and composites, due to attenuation effects and strong scattering.
That shortcoming mostly depends on ultrasound pulse shape, which one can be improved by using a different type of generation. Usage of laser ultrasound generation becoming more popular, in virtue of much higher accuracy. The special optoacoustic generator is used in the Contact laser ultrasound method to generate ultrashort ultrasonic pulses. At the same peak frequencies of ultrasound pulses, the duration of a laser ultrasound pulse is 6–7 times shorter (0.07 μs) than for piezo ultrasound (0.58 μs) at the same central frequencies. The short duration of the probing ultrasonic pulse provides increased spatial resolution. Due to this, the accuracy of determining the depth of a defect and its detection in laser ultrasound is 6-7 times higher than that of traditional ultrasound. Record scattered signals provided by a broadband piezoelectric transducer.
“Line 297-298: How the adhesive properties were measured?”
COMMENTS:
We did not measure adhesive properties, in connection with your comment, we have to change the sentence from:
“Better tensile properties of the overlap joint were caused by the summation of both types of joint strength (…)”
to:
“Better tensile properties of the overlap joint could be caused by the summation of both types of joint strength (…)”
and yellow-highlighted it.
“The discussion could be more solid. So the results should be discussed with previously published literature”
COMMENTS:
We have made some additional discussion basing on your advice.
“Also, revise the manuscript carefully to reduce the typographical and linguistic errors in the manuscript”
COMMENTS: We have followed the advice and removed all spotted language issues.
Reviewer 2 Report
The paper entitled "The examination of restrained joints created in the process of multi-material FFF additive manufacturing technology” falls within the scope of the Materials Journal and shows technical relevance.
In this paper the authors present an interesting study, the work focuses on experimental production and non-destructive testing of restrained joints of modified PLA (polylactic acid) and ABS (Acrylonitrile butadiene styrene) 3Dprinted on RepRap 3D device that works on the "open source" principle.
The paper is very well written and presented and the material is publishable, but requires improvement. In this sense, there are some suggestions on the attached paper that should be addressed before publishing.
Suggestion 01
The limitations of the study should also be included in the introduction section.
Suggestion 02
In the description of objectives at the end of the introduction section, background of the laser non-destructive measurement procedure should have been described. References about previous research applications of this method should also be added.
Suggestion 03
Novelty unclear: What is the original contribution of the study? The methodology is not very enlightening on the subject. Novelty should be made as clear as possible.
Suggestion 04
In figure 9 and in the images in table 1 the optical degree of magnification has not been specified. Furthermore, in Figure 9, some measurements of the void imperfections at the end of the wavy borderline of the joint are missing and it is not specified whether the images in table 1 have been obtained with the same confocal microscope or not.
Suggestion 05
What has been the procedure used to obtain the images of figures 11, 12 and 13? What is the degree of magnification?
Suggestion 06
Main data found in the results must be presented in the conclusion section in order to strengthen the proposed joint method.
Author Response
Revision #2
Dear Reviewer,
We would like to thank you for your revision and valuable comments.
In connection with your suggestion we have made the following corrections:
“ The limitations of the study should also be included in the introduction section.”
COMMENTS:
We put an additional paragraph about limitations and green-highlighted it:
Mentioned materials have their limitations. PLA is characterized by a very low plastification temperature and a lack of UV resistance. ABS has low moisture resistance, and additional in 3D printing solutions it has a significant shrinkage which makes it difficult to use – on manufactured parts could appear cracks and delaminations. Regarding the above-mentioned restrictions, it is necessary to provide the best possible additive manufacturing technology and select proper materials for their usage conditions.
“In the description of objectives at the end of the introduction section, the background of the laser non-destructive measurement procedure should have been described. References about previous research applications of this method should also be added.”
COMMENTS:
Laser ultrasonic structuroscopy, used in this study, is similar to traditional ultrasonic systems operating in echo-pulse mode [27, 28]. The generation of short ultrasonic pulses of strictly controlled form occurs in an optoacoustic cell [29]. Figure 7 shows a schematic diagram of this cell for the diagnosis of materials with one-way access. A pulse generated by an Nd: YAG laser is transmitted to the front side of a special optical-acoustic generator (OAG) through a fiber optic cable, an optical beamforming system, and a transparent prism. OAG is a plane-parallel plate made of a special plastic that absorbs light [30]. The transparent prism is in acoustic contact with the OAS, being at the same time a sound-conducting channel of a broadband piezoelectric transducer. Access on the one hand and acoustic contact are provided by pressing the OAG plane to the front of the object with a thin layer of contact liquid [31].
Dvořák, P.; Štoller, J.; Baláž, T.; Krejčí, J. Nondestructive Testing of Critical Infrastructure Objects. Transport Means, Kaunas, Lithuania. 2018, 855-859 Dvořák, P.; Štoller, J. Ultrasound Diagnosis of Protective Structures after Contact Explosion. Transport Means. Kaunas, Lithuania. 2014, 264-267 Štoller, J.; Zezulova, E. Use of ultrasound-The ultrasonic pulse velocity method for the diagnosis of protective structures after the load of TNT explosion, ICMT. 2017, 230-235 Svoboda, P.; Kravcov, A.; Pospíchal, V.; Morozov, N.; Zezulová, E. Quality assessment of bored pile foundations by a set of non-destructive testing methods, 2019, 87-91 Kravcov, A.; Shibaev, I. Examination of structural members of aerial vehicles by laser ultrasonic structuroscopy, International Journal of Civil Engineering and Technology, vol. 9, issue 11. 11/2018 , 2258-2265 Kravcov, A.; Platek, P.; Pospichal, V.; Koperski, W.; Internal structure research of 3D printed cellular structures by laser-ultrasonic structuroscopy, ICMT. 2019, 92-99
“Novelty unclear: What is the original contribution of the study? The methodology is not very enlightening on the subject. Novelty should be made as clear as possible.”
COMMENTS:
We have made additional background revision and add a paragraph about the usage of suggested technology:
Basing on the last five-years research work, the most significant articles groups connected with plastics joints are focused on composites with the addition of polymers [21-25]. A significant part is taking into account conventional material joining type. More advanced technologies were shown in Eslami et al. research [25], where it had been used friction stir welding technology of polymers. A small amount of that kind of research and lack of new solutions for plastics was the main background to investigate a patented technology (Patent application number: P432635) and made tests which were shown in this article.
In this work, the effectiveness and quality of the joints using two materials were taken into account. To reach very good joint mechanical properties it had been designed a proper additive manufacturing FFF process and special joint geometry. The prepared joints were tested using a non-destructive laser amplified ultrasonic method, supplemented by the microscopic observations of the joints before and after tensile tests. The research allowed for obtaining the mixed material properties which could be used in many different industrial applications. Additionally, investigated polymers joining technology is a real alternative for conventional joining techniques (glue connections, screw connections, etc.). The research allows to state, that developed connection geometry and manufacturing technology could be helpful to reach polymer joints characterized by good mechanical properties.
“In figure 9 and in the images in table 1 the optical degree of magnification has not been specified. Furthermore, in Figure 9, some measurements of the void imperfections at the end of the wavy borderline of the joint are missing and it is not specified whether the images in table 1 have been obtained with the same confocal microscope or not.”
COMMENTS:
We put scales in all images to clarify the magnification issue.
Measurements of some porous areas were not able to carry out, because of its irregular shape and stochastic nature of its presence in the material structure. Porosity shown in figure 9 (after another reviewer advice – figure 10) was shown mainly to illustrate the nature of that kind of imperfection.
In the sentence concerning table 1, we put additional information that images in it were taken using an Olympus LEXT 4100 confocal microscope.
“What has been the procedure used to obtain the images of figures 11, 12 and 13? What is the degree of magnification?”
COMMENTS:
At the beginning of chapter 5, we put an additional sentence: “Additional microscopic observations (shown in figures 12,13 and 14) were made using the Olympus LEXT 4100 confocal microscope.
As it was mentioned before, we put scales in all microscopic images.
“Main data found in the results must be presented in the conclusion section in order to strengthen the proposed joint method.”
COMMENTS:
We have made some additional discussion basing on your advice.
We put additional conclusions connected with the obtained results.
Round 2
Reviewer 1 Report
Revision were adequately made based on the comments and the manuscript now can be considered for publication.
Reviewer 2 Report
Once the initial assessments have been taken into account, this reviewer considers the paper revised version appropriate for publication.